# Effects of 2-Phenylethanol on Controlling the Development of *Fusarium graminearum* in Wheat

**DOI:** 10.3390/microorganisms11122954

**Published:** 2023-12-10

**Authors:** Shufang Sun, Nawen Tang, Kun Han, Qunqing Wang, Qian Xu

**Affiliations:** 1National Key Laboratory of Wheat Improvement, College of Agronomy, Shandong Agricultural University, Taian 271018, China; sun17653475426@163.com (S.S.); tangnawen95@163.com (N.T.); 2Departmen of Plant Pathology, College of Plant Protection, Shandong Agricultural University, Taian 271018, China; hkun1216@163.com

**Keywords:** 2-phenylethanol, *Fusarium graminearum*, volatile organic compounds (VOCs), antifungal mechanism, metabolomics

## Abstract

Applying plant-derived fungicides is a safe and sustainable way to control wheat scab. In this study, volatile organic compounds (VOCs) of wheat cultivars with and without the resistance gene *Fhb1* were analyzed by GC-MS, and 2-phenylethanol was screened out. The biocontrol function of 2-phenylethanol on *Fusarium graminearum* was evaluated in vitro and in vivo. Metabolomics analysis indicated that 2-phenylethanol altered the amino acid pathways of *F. graminearum*, affecting its normal life activities. Under SEM and TEM observation, the mycelial morphology changed, and the integrity of the cell membrane was destroyed. Furthermore, 2-phenylethanol could inhibit the production of mycotoxins (DON, 3-ADON, 15-ADON) by *F. graminearum* and reduce grain contamination. This research provides new ideas for green prevention and control of wheat FHB in the field.

## 1. Introduction

*Fusarium* head blight (FHB), caused by *Fusarium graminearum*, is a devastating disease of wheat (*Triticum aestivum*) [1]. FHB has a severe impact on wheat yield, generally leading to a 10% to 40% reduction in yield and even no harvest in severe cases [2]. In addition to threatening yield and reducing the quality of wheat, mycotoxins (Deoxynivalenol, 3-Acetyldeoxynivalenol, 15-Acetyldeoxynivalenol, etc.) produced by *F. graminearum* also pollute the grain and harm human and animal health [3]. To date, research efforts include digging for genes relevant to FHB, breeding disease-resistant varieties, and developing fungicides to control scab. Wheat scab resistance is a quantitative trait controlled by multiple genes, and *Fhb1*-*Fhb7* are the resistance genes that have been identified and named [4,5,6,7]. *Fhb1*, a major gene for resistance to Fusarium head blight (FHB), has been widely used in wheat breeding worldwide for improving FHB resistance [8,9]. Given the limited disease-resistant germplasm, fungicides are the primary strategy for controlling FHB.

In agricultural production, fungicides play a crucial role in food security. Conventional fungicides are mainly chemical fungicides, such as carbendazim, tebuconazole, metconazole, etc. [10,11]. Single and repeated use of chemical fungicides leads to drug resistance, environmental pollution, toxic residues, and grain quality deterioration [12,13]. With the increasing awareness of environmental protection among the public, biological control is gradually receiving attention. Thus, there is an urgent need to seek a highly efficient, eco-friendly, non-toxic fungicide. Plant-derived fungicides, as an effective solution for the control of wheat scab, are characterized by low toxicity, good specificity, biodegradability, and environmental friendliness [14].

Plant-derived small-molecule volatile organic compounds (VOCs), produced by plant secondary metabolism, can effectively inhibit the growth and proliferation of various plant pathogens [15,16]. VOCs are widely applied to food preservation and controlling plant diseases and pests [17,18,19]. 2-Phenylethanol is a flavor compound that can be isolated from flowering plants (e.g., geranium, petunias, neroli) [20] and a variety of yeast species (e.g., *Aspergillus oryzae*, *Yarrowia lipolytica*, *Saccharomyces cerevisiae*) [21,22]. It has wide applications in the perfume, cosmetics, pharmaceutical, and food industries [23,24]. Numerous studies have shown that 2-phenylethanol could inhibit bacteria (e.g., *Escherichia coli*, *Enterococcus faecium*) [25,26] and *Penicillium* spp. (e.g., *Penicillium expansum*, *Penicillium nordicum*) [27]. However, little attention has been paid to the antimicrobial activity of 2-phenylethanol against plant pathogens.

In this paper, we detected 2-phenylethanol by GC-MS in FHB-resistant wheat inoculated with *F. graminearum*. Subsequently, the antimicrobial activity of 2-phenylethanol against *F. graminearum* was evaluated in vitro/vivo. Then, we examined the mechanisms of how 2-phenylethanol inhibited the growth of *F. graminearum* using scanning electron microscopy (SEM), transmission electron microscopy (TEM), metabolomics analysis, and measurement of the levels of deoxynivalenol (DON) and its derivatives. Lastly, we explored the relationship between the length of the carbon chain and the antifungal activity of alcohols.

## 2. Materials and Methods

### 2.1. Fungicides and Plants

*F. graminearum* (PH-1) was provided by the Shandong Province National Key Laboratory of Wheat Improvement (Shandong Agricultural University, Taian, China). *F. graminearum* was cultured in PDA (potato dextrose agar) or CMC (carboxymethylcellulose sodium) medium at 25 °C. The PDA (potato dextrose agar) medium contained 200 g of potato, 15 g of dextrose, and 12 g of agar in 1 L of water. The CMC (carboxymethylcellulose sodium) medium contained 200 g of CMC, 2 g of NaNO_3_, 1 g of KNO_3_, 0.5 g of MgSO_4_·7H_2_O, and 1 g of yeast extract in 1 L of water.

Apogee73S2 (+*Fhb1*, FHB resistant variety) and Apogee (−*Fhb1*, FHB susceptible variety) were provided by Dr. D. Garvin of the University of Minnesota.

### 2.2. Chemicals

The standards of DON, 3ADON, and 15ADON were purchased from Yuanye Bio-Technology Co., Ltd. (Shanghai, China). CMC medium was purchased from Aladdin Biochemical Technology Co., Ltd. (Shanghai, China).

All chemical reagents were purchased from Tianjin Kaitong Chemical Reagent Co., Ltd. (Tianjin, China) or Tedia Co, Inc (Fairfield, OH, USA).

### 2.3. Gas Chromatography–Mass Spectrometry (GC-MS)

The wheat spikes (Apogee73S2 and Apogee) inoculated with *F. graminearum* and water were placed in headspace bottles, respectively. Every treatment had three biological replicates. Chromatography was performed on a GC-MS-TQ8040 system (Shimadzu, Kyoto, Japan) equipped with an AOC-6000 Plus multifunctional autosampler and mass spectrometer [28]. Solid-phase microextraction (SPME) was performed using an AOC-6000 multifunctional automatic sampler and GC-MS with the following standard SPME parameters: SPME fiber, FIB-C-WR-95/10; ageing temperature, 240 °C; ageing time before extraction, 30 min; equilibration temperature, 40 °C; equilibration time, 5 min; extraction time, 30 min; injection port temperature, 250 °C; desorption time, 2 min; and ageing time after extraction, 5 min. Chromatographic separation of metabolites was performed with an inert cap pure-wax column (30 m × 0.25 mm × 0.25 m). For the GC-MS analysis, the initial oven temperature was set at 50 °C, followed by a gradient of 10 °C/min up to 250 °C and held for 10 min. The following further parameters were set in the mass spectrometer: carrier gas pressure, 83.5 kPa; injection mode, split; split ratio, 5:1; ion-source temperature, 200 °C; interface temperature, 250 °C; detector voltage, tuning voltage +0.3 kV; and acquisition mode, MRM.

### 2.4. In Vitro Fungicidal Assay

For the median effective 2-phenylethanol concentration (EC_50_) experiment, this was determined as described in [29]. Mycelial plugs (6 mm diameter) of 3-day-old colonies were placed at the center of PDA media containing 2-phenylethanol (0, 0.05, 0.1, 0.2, 0.4, 0.6, 0.8, 1.0, or 1.2 mg/mL). After incubating in darkness at 25 °C for 3 days, the diameter of the colonies was recorded. Each treatment was repeated three times. The growth inhibition rate (%) was calculated using the following formula:$$\mathrm{Inhibition}\;\mathrm{rate}\;\left(\%\right)=\frac{\left(\mathrm{Growth}\;\mathrm{diameter}\;\mathrm{in}\;\mathrm{control}-\mathrm{Growth}\;\mathrm{diameter}\;\mathrm{in}\;\mathrm{treatment}\right)}{\mathrm{Growth}\;\mathrm{diameter}\;\mathrm{in}\;\mathrm{control}\;}\times 100$$

EC_50_ was calculated using SPSS software version 26.0.

### 2.5. In Vivo Fungicidal Assay

Six mycelial plugs (6 mm in diameter) of *F. graminearum* (PH-1) were taken from the margin of a 3-day-old colony and added to CMC medium. After culturing on a shaker for 3 days (25 °C, 220 rpm), the spore suspension was filtered using gauze. The spores were re-suspended in distilled water and counted using a hemocytometer under an optical microscope.

Field efficacy experiments of 2-phenylethanol against *F. graminearum* on wheat spikes were conducted in Taian, China. At the anthesis stage, the spikes were inoculated with 10 µL spore suspension (5 × 10^5^ spores/mL) of *F. graminearum*. The wheat spikes (Fielder, susceptible to FHB) were evenly sprayed with 2-phenylethanol diluted in water to different final concentrations (0.6, 1.2, or 2.4 mg/mL). The same amount of water was set as a negative control (CK) and tebuconazole (215 μg/mL) as a positive control (CK1). Each treatment was performed three times. The number of diseased spikelets and pathogenic phenotypes were recorded 14 days later. The percentage of diseased spikelets was calculated as follows:$$\mathrm{The}\mathrm{percentage}\mathrm{of}\mathrm{diseased}\mathrm{spikelets}\left(\%\right)=\frac{\mathrm{Diseased}\mathrm{spikelets}}{\mathrm{Total}\mathrm{number}\mathrm{of}\mathrm{spikelets}}\times 100$$

### 2.6. Metabolomics Analysis

#### 2.6.1. Metabolite Extraction

Metabolomics analysis was performed as described in Zhao et al. [30], with minor modifications. A 0.1 g mycelial sample was weighed and mixed with 1.5 mL extraction solution (methanol:water:formic acid, 2:1:1, *v*/*v*/*v*). The mixture was subjected to ultrasound treatment at room temperature for 20 min and then rotated at 4 °C for 12 h. Subsequently, it was centrifuged at 13,000 rpm for 5 min, and 2 mL of the supernatant was transferred to a new centrifuge tube. The filtrate was concentrated using a rotary evaporator, re-dissolved in 100 μL of 50% methanol, and filtered through a syringe filter (0.22 μm) for analysis preparation. Each treatment consisted of three biological replicates.

#### 2.6.2. UHPLC-Q-Exactive Orbitrap MS (UPLC-QE-MS) Analysis

Extracts were analyzed with a Q-Exactive quadrupole-Orbitrap mass spectrometer equipped with a heated electrospray ionization (ESI) source (Thermo Fisher Scientific, Bremen, Germany). Chromatographic separation of metabolites was performed with a Thermo HYPERSIL GOLD C18 chromatographic column (2.1 × 100, 1.9 µm), and the column temperature was 35 °C. The solvent gradient was set as follows: A, 0.1% of acetic acid in distilled water; B, 0.1% of acetic acid in acetonitrile. The solvent gradient was set as follows: 0–2.5 min, 98–95% A; 2.5–7.5 min, 95–70% A; 7.5–10.5 min, 70–2% A; 10.5–13 min, 2% A, 13.0-13.1 min, 2–98% A; 13.1–11.5 min, 98% A; injection volume, 3 µL.

The working conditions of the mass spectrometer were as follows:

In positive polarity mode, spray voltage, 3.8 kv; sheath gas, 40; pilot gas, 10; capillary temperature, 350 °C; resolution, 17,500; microsweep, 1; AGC target, 2 × 10^5^; normalized collision energy, 50. In negative polarity mode, spray voltage, 2.9 kv; sheath gas, 40; pilot gas, 0; capillary temperature, 350 °C; resolution, 17,500; microsweep, 1; AGC, target 2 × 10^5^; normalized collision energy, 50.

#### 2.6.3. Metabolomics Data Analysis

Raw metabolomics data detected using the UPLC-QE-MS were analyzed using Compound Discoverer software 3.3. The annotated name, retention time, peak area, and precise molecular weight of the metabolite were obtained from Compound Discoverer 3.0.

### 2.7. Determination of DON, 3ADON, 15ADON Content

At the anthesis stage, wheat spikes were inoculated with *F. graminearum* using the single-flower drop injection method [31]. The spore suspension was 5 × 10^5^ spores mL^−1^. Meanwhile, the spikes were sprayed with distilled water or 2-phenylethanol (1.2 mg/mL). Each treatment had six biological replicates. After 14 days, the spikes were collected and stored at −80 °C.

DON, ADON, and 15ADON were extracted from fresh spikes following the method described by Han et al. [32] with minor modifications. The spikes were ground in liquid nitrogen. Next, 100 mg frozen powder of each sample was weighed and 1.5 mL of 75% methanol was added. Samples were subjected to ultrasound at room temperature for 30 min and centrifuged at 4 °C, 13,000 rpm for 5 min. Next, the supernatant was transferred to a new centrifuge tube and vacuum-concentrated to constant weight. Finally, dried samples were redissolved with 100 µL of 20% acetonitrile and filtered with a 0.22 µm filter membrane. The DON, ADON, and 15ADON content were separated and quantified using an ultra-efficient liquid-phase triple quadrupole mass spectrometer (TSQ02-10001, Thermo Fisher, Waltham, MA, USA).

### 2.8. Morphology and Ultrastructure of Fungal Hyphae

Three mycelial plugs (6 mm in diameter) of *F. graminearum* were inoculated in the center of a PDA medium amended with compound 2-phenylethanol at the EC_50_ (0.328 mg/mL) value. A PDA medium without 2-phenylethanol treatment was used as a control. They were incubated in an incubator for 3 days at 25 °C with 24 h dark. After 3 days, mycelia were collected and washed three times with PBS buffer (PH = 7). The experiment was repeated three times for each concentration.

#### 2.8.1. SEM Observation

According to the method in [33], SEM was carried out to determine the effect of 2-phenylethanol on the mycelial morphology of *F. graminearum*. The mycelia were fixed in 2.5% glutaraldehyde at 4 °C for 12 h and washed with PBS buffer (PH = 7.0). The hyphae samples were dehydrated with 5%, 55%, 75%, 85%, 95%, and 100% ethanol for 30 min. Subsequently, fungal samples were dried at the critical point of CO_2_ and coated with gold. Finally, the resultant mycelium morphology was photographed using SEM (Zeiss, Jena, Germany).

#### 2.8.2. TEM Observation

For TEM observation, the mycelia were successively soaked in the 2.5% glutaraldehyde for 12 h and 1% OsO_4_ solution for 2 h. Samples were dehydrated with gradient ethanol (5%, 55%, 75%, 85%, 95%, or 100%) and embedded in resin for ultrathin sectioning. Sections were double-stained using uranyl acetate and lead citrate. The cell ultrastructure of *F. graminearum* was examined and photographed with TEM (JEOL, Tokyo, Japan) [34].

### 2.9. Antifungal Activity of Benzyl Alcohol, 2-Phenylethanol, and 3-Phenylpropanol

The mycelial plugs (6 mm in diameter) of *F. graminearum* (PH-1) were transferred from a 3-old-day PDA medium into a new PDA medium containing benzyl alcohol, 2-phenylethanol, and phenylpropanol (0, 0.2, 0.4, 0.6, 0.8, 1.0, 1.2, 1.4 mg/mL). The PDA plates were placed in an incubator for three days (25 °C). The growth phenotypes and diameters of the mycelium were recorded after three days.

### 2.10. Statistical Analysis

The data were analyzed using one-way ANOVA in IBM SPSS Statistics 22.0 software (SPSS Inc., Chicago, IL, USA). *p* < 0.05 indicated a significant difference. Figures were plotted using SigmaPlot 14.0 (Systat Software, Inc., San Jose, CA, USA, Sigma Plot).

## 3. Results

### 3.1. Detection of the 2-Phenylethanol in Wheat Spikes

To search for plant-derived antifungal compounds against FHB, we analyzed the spike VOCs produced during incompatible wheat–*F. graminearum* interactions. Secondary mass spectrometry is commonly used for qualitative analysis of substances. Based on secondary mass spectrometry fragment ions, we matched a volatile organic compound, 2-phenylethanol (122, 91, 65, Figure 1E). The results showed that the 2-phenylethanol content in the spike volatiles of Apogee73S2(w/*Fhb1*) was significantly higher than that of Apogee (w/o resistant gene) (Figure 1A–D), indicating that 2-phenylethanol contributes to the defense response of wheat to *F. graminearum*.

### 3.2. 2-Phenylethanol Inhibits the Growth of F. graminearum In Vitro

2-Phenylethanol exhibited excellent inhibition of the mycelial growth of *F. graminearum* with the EC_50_ value at 0.328 mg/mL (Figure 2A). Compared to the control, the antifungal rates of 2-phenylethanol were only 20.44% and 32.44% at concentrations of 0.05 and 0.1 mg/mL, respectively. As the concentration of 2-phenylethanol increased, its inhibitory effect on the growth of *F. graminearum* became more significant. When the concentration of 2-phenylethanol reached 1.2 mg/mL, the inhibitory rate was 100%. The toxicity regression equation of 2-phenylethanol against *F. graminearum* was y = 2.20x − 0.071 (*R*^2^ = 0.970).

### 3.3. Effect of 2-Phenylethanol on F. graminearum Growth In Vivo

The field control effectiveness of 2-phenylethanol against *F. graminearum* was evaluated based on concentration screening in vitro experiments. In order to evaluate 2-phenylethanol’s field control effect on FHB, the spikes were co-treated with *F. graminearum* and 2-phenylethanol. The concentration of 2-phenylethanol at 0.6 and 1.2 mg/mL significantly reduced diseased lesions of spikes (Figure 2B–D). At concentrations of 0, 0.6, 1.2, and 2.4 mg/mL, the average incidence of diseased spikelets per ear of wheat was 58.09%, 31.77%, 19.5%, and 12.38%, respectively. The 2-phenylethanol at a concentration of 2.4 mg/mL was comparable to the chemical fungicide tebuconazole in controlling scab (Figure 2E,F). The number of diseased spikelets was counted, and it was discovered that spraying 2-phenylethanol at a concentration of 2.4 mg/mL reduced the rate of diseased spikelets by 78% compared to the control (Figure 2G).

### 3.4. Metabolomics Analysis of F. graminearum under 2-Phenylethanol Stress

The principal component analysis (PCA) of metabolites across all samples showed that the first two principal components explained 56.78 and 13.96% of the total variance, respectively. The six replicates of the same group could be clustered together well, and the two groups of samples were significantly separated, which indicated significant changes in metabolites of *F. graminearum* treated with 2-phenylethanol (Figure 3A).

A total of 33 annotated metabolites were identified in the comparison of treated and untreated samples (Log_2_Fold Change > 1 and <−1, *p*-value < 0.05. The identified metabolites were amino acids and derivatives, fatty acids and derivatives, organic acids, etc. (Figure 3B). Eight metabolites were up-regulated, and twenty-five metabolites were down-regulated (Figure 3C). DON (deoxynivalenol) is a characteristic secondary metabolite produced by Fusarium species. 15-Acetyldeoxynivalenol (15ADON) is an acetylated product of DON, which is used as an important virulence factor [35]. Compared with the CK group, the content of 15ADON in the treatment group decreased.

To further analyze the antifungal mechanism of 2-phenylethanol, we performed a KEGG pathway analysis to identify the differential metabolites involved in metabolism pathways. There were significant changes in seven metabolic pathways (*p* < 0.05), including aminoacyl-tRNA biosynthesis, biosynthesis of unsaturated fatty acids, cysteine and methionine metabolism, lysine degradation, metabolic pathways, biosynthesis of secondary metabolites, and arginine and proline metabolism (Figure 3D). These results demonstrated that 2-phenylethanol dramatically changed the amino acid metabolism pathway of *F. graminearum*.

### 3.5. 2-Phenylethanol Inhibited the Production of DON Toxin by F. graminearum

Due to the limitations of the extraction method for metabolomics, this experiment further targeted the determination of DON and its derivatives. Wheat spikes sprayed with 2-phenylethanol had a lower content of the three DON toxins than the control (Figure 4). This result indicated that 2-phenylethanol inhibited the production of DON toxin by *F. graminearum*.

### 3.6. 2-Phenylethanol Changed the Ultrastructure of F. graminearum Mycelia

From the SEM images, mycelia in the control group were smooth and regular (Figure 5A). However, the surface of mycelia incubated with 2-phenylethanol were shrunken and wrinkled (Figure 5D). Under TEM, the cell organs of normal mycelia were integral and evenly distributed in the cytoplasm (Figure 5B,C). Meanwhile, the boundary between organelles was visible. By contrast, plasma membranes of mycelia exposed to 2-phenylethanol were destroyed, and the contents of the organelles leaked (Figure 5E,F).

### 3.7. The Relationship between the Antifungal Activity and Length of the Carbon Chain

To better excavate the potential alcoholic fungicides, we examined the relationship between carbon chain length and antifungal activity. Figure 6 displays preliminary results of three alcoholic compounds against the growth of *F. graminearum*. The concentrations of benzyl alcohol, 2-phenylethanol, and 3-phenylpropanol that completely inhibited the growth of *F. graminearum* were 1.4, 1.2, and 1.0 mg/mL, respectively. The EC_50_ values of three alcoholic compounds were different: benzyl alcohol > 2-phenylethanol > 3-phenylpropanol (Table 1). The three compounds have different carbon chain lengths in their structures, with longer ones resulting in higher antifungal activity.

## 4. Discussion

VOCs mainly refer to phenols, alcohols, aldehydes, esters, organic acids, and other substances [16,36]. Studies have shown that many VOCs have antimicrobial activity. The volatile emitted by Trichoderma could effectively inhibit the pathogenic strains of potato late blight [37]. Linalool, produced by strawberries, down-regulated the expression of rate-limiting enzymes in the ergosterol biosynthesis pathway and reduced the infection of Botrytis cinerea [38]. The nonanal released by lima beans up-regulated the expression of the PR gene, resulting in reduced lesion area caused by *Pseudomonas syringae pv. Syringae* [39]. According to previous reports, microbial infection significantly altered the emission of plant VOCs [40]. In this study, we found that infection by *F. graminearum* up-regulated 2-phenylethanol production by wheat.

2-Phenylethanol, a natural product, has fungicidal properties and can be a candidate for green fungicide precursors. Prior studies showed that 2-phenylethanol could control postharvest diseases such as *Phytophthora infestans*, *Botrytis cinerea*, and *Penicillium molds*, which is important for preserving the freshness of fruits and vegetables like potatoes, strawberries, and citrus fruits after harvest [41,42,43]. In particular, 2-phenylethanol can also be used in humans due to its ability to inhibit *Candida* species growth [44]. In this study, 2-phenylethanol was detected by GC-MS after inoculation with *F. graminearum* in Apogee73S2 with the FHB resistance gene *Fhb1*. In vivo and in vitro tests demonstrated that 2-phenylethanol inhibited the growth of *F. graminearum* (Figure 2). Therefore, the mechanism of 2-phenylethanol against *F. graminearum* growth was further explored in this paper.

Amino acids are essential for life activities and are the basic structural units of proteins. Metabolomics analysis illustrated that differential metabolites of mycelia exposed to 2-phenylethanol were mostly amino acids compared with those in the control group (Figure 3C). Jones et al. [45] identified arginine as a precursor of nitric oxide, and it is crucial for the germination of conidia in *Blumeria graminis* and *Colletotrichum coccodes*. The synthesis of lysine is necessary for the production of proteins and cell walls of bacterial peptidoglycan [46]. KEGG analysis revealed that the aminoacyl-tRNA biosynthetic pathway was significantly altered in *F. graminearum* treated with 2-phenylethanol (Figure 3C). The function of aminoacyl-tRNA biosynthesis is to accurately match amino acids with tRNAs containing corresponding anticodons [47]. The results suggested that 2-phenylethanol interfered with the protein synthesis/degradation of *F. graminearum* and affected its normal growth. Additionally, amino acids play a crucial role in the synthesis of DON toxins in *F. graminearum*. L-isoleucine, a branched-chain amino acid, plays three crucial roles in the production of DON toxins in *F. graminearum*: firstly, it serves as a precursor for the synthesis of DON toxins; secondly, as a co-factor for enzymes, it enhances the activity of enzymes involved in DON synthesis; and finally, it regulates metabolic pathways, influencing the production and utilization of other metabolites [48,49,50]. Consistent with these results, both non-targeted and targeted metabolomic analyses indicated a significant inhibition of DON toxin synthesis and a reduction in the accumulation of DON toxins (Figure 3 and Figure 4). The inhibition of DON toxin synthesis in *F. graminearum* by 2-phenylethanol is of significant importance in ensuring human health and enhancing wheat yield and quality.

In addition, the unsaturated fatty acid biosynthetic pathway of *F. graminearum* treated with 2-phenylethanol underwent significant changes (Figure 3D). Fatty acids are essential for maintaining normal cell activity, such as energy storage, integrity and dynamics of biological membranes, cellular metabolism, and cellular signaling [51]. Changes in the pathway of unsaturated fatty acid biosynthesis can result in alterations in the permeability and fluidity of cell membranes, which is regarded as a response mechanism when cells face stress [52]. The results obtained from electron microscopy in our study were consistent with those described above. The cell membranes of the mycelia incubated with 2-phenylethanol exhibited shrinkage and other changes, while the membranes of the various organelles were severely damaged (Figure 5).

After confirming the antifungal activity of 2-phenylethanol, we further explored the relationship between the structure of alcohols and their antifungal activity. The antifungal activity of alcohols (benzylalcohol, 2-phenylethanol, and 3-phenylpropanol) is positively correlated with the length of the carbon chain. By comparing the antifungal activity of benzylalcohol, 2-phenylethanol, and 3-phenylpropanol, as well as the production cost, we finally determined that the most suitable fungicide for application in field production is 2-phenylethanol.

Above all, this study detected 2-phenylethanol by GC-MS and demonstrated its in vitro and in vivo antifungal activity against *F. graminearum*. The antifungal mechanism of 2-phenylethanol was to hinder protein synthesis, disrupt the integrity of cell membrane, and affect the normal growth of *F. graminearum* (Figure 7). Additionally, it was found to effectively reduce the production of DON toxins (DON, 3ADON, 15ADON). 2-Phenylethanol is an ideal plant-derived fungicide for controlling wheat scab caused by *F.graminearum*.

## Figures and Tables

**Figure 1 microorganisms-11-02954-f001:**
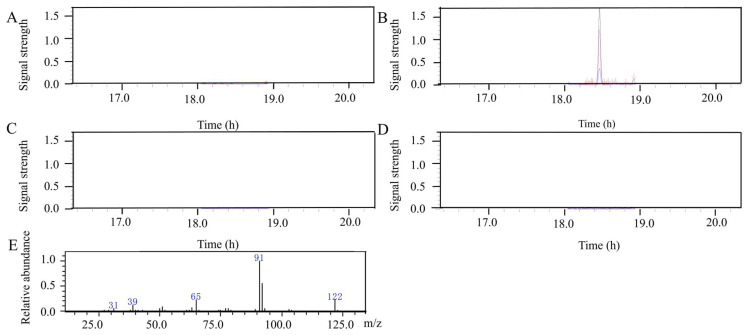
Chromatograms of GC-MS. (**A**) FHB-resistant variety (Apogee73S2) inoculated with water. (**B**) FHB-resistant variety (Apogee73S2) inoculated with *F. graminearum*. (**C**) FHB-susceptible variety (Apogee) inoculated with water. (**D**) FHB-susceptible variety (Apogee) inoculated with *F. graminearum*. (**E**) The secondary mass spectrogram of 2-phenylethanol. *m/z* represents mass-to-charge ratio. Relative abundance refers to the strength of the ion signal corresponding to each *m/z* ratio. The black lines refer to qualitative ions; the red lines and blue lines refer to quota ions. The arrow indicates 2-phenylethanol.

**Figure 2 microorganisms-11-02954-f002:**
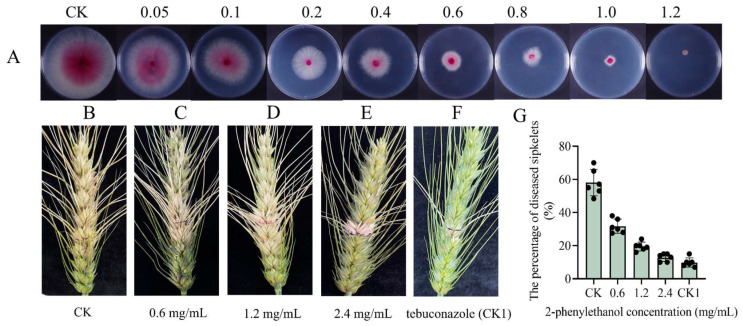
In vitro and in vivo antifungal activities of 2-phenylethanol against *F. graminearum*. (**A**) Mycelia growth of *F. graminearum* on PDA medium containing different concentrations of 2-phenylethanol (at 0, 0.05, 0.1, 0.2, 0.4, 0.6, 0.8, 1.0, 1.2 mg/mL). (**B**–**F**) Protective activities of 2-phenylethanol on wheat spikes infected by *F. graminearum* (PH-1) after 14 days. Spikes treated with sterilized distilled water (CK); spikes treated with 2-phenylethanol (at 0, 0.6, 1.2, 1.4 mg/mL); spikes treated with tebuconazole (CK1). (**G**) Reduction in the disease severity of spikes treated with 2-phenylethanol for 14 days. The data are mean ± standard error (*n* = 6).

**Figure 3 microorganisms-11-02954-f003:**
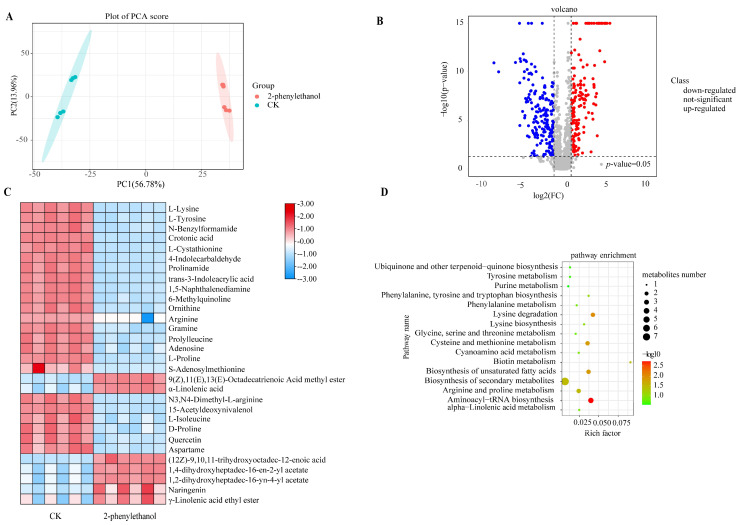
Metabolomics analysis of *F. graminearum* between control group (CK) and treatment group (2-phenylethanol). (**A**) Principal component analysis (PCA) of samples in CK and treatment (2-phenylethanol) groups based on metabolic profiles. (**B**) Volcano plots of all up- and down-regulated metabolites. (**C**) Heatmap exhibits all differential annotated metabolites detected from mycelia treated with 2-phenylethanol (0.328 mg/mL) compared with CK. Red, blue, and gray dots represent metabolites with high abundance, low abundance, and no significant difference, respectively. (**D**) KEGG pathway enrichment analysis of all differential metabolites between control group (CK) and experimental group (2-phenylethanol). The color scale showed the values (log2 fold change). The horizontal coordinate-axis X indicates the *p*-value. The vertical coordinate-axis Y indicates multiple biological processes. The smaller the *p*-value, the redder the bubble.

**Figure 4 microorganisms-11-02954-f004:**
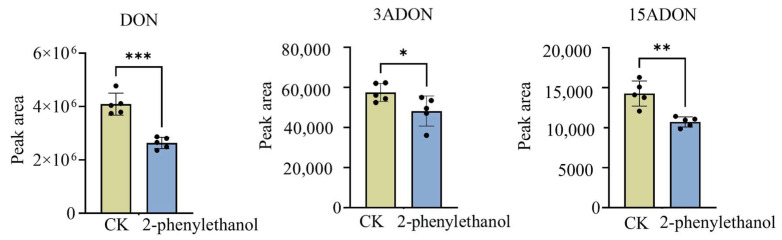
DON, 3ADON, and 15ADON accumulation of wheat spikes after inoculation with *F. graminearum* for 14 days. CK group: spikes treated with distilled water; 2-phenylethanol: spikes treated with 2-phenylethanol (1.2 mg/mL). Data are presented as mean ± s.d. of *n* = 6 biologically independent samples. The asterisk (*) above the columns indicates a statistically significant difference (* indicates *p* < 0.05; ** indicates *p* < 0.01; *** indicates *p* < 0.001).

**Figure 5 microorganisms-11-02954-f005:**
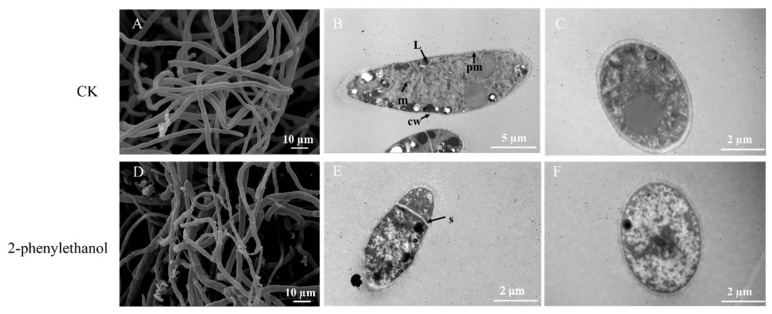
SEM and TEM images of *F. graminearum* cellular structure. (**A**–**C**) Mycelia not treated with 2-phenylethanol. (**D**–**F**) Mycelia treated with 2-phenylethanol at 0.328 mg/mL (EC_50_). L, lipid droplet; pm, plasma membrane; cw, cell wall; m, mitochondria; s, septum.

**Figure 6 microorganisms-11-02954-f006:**
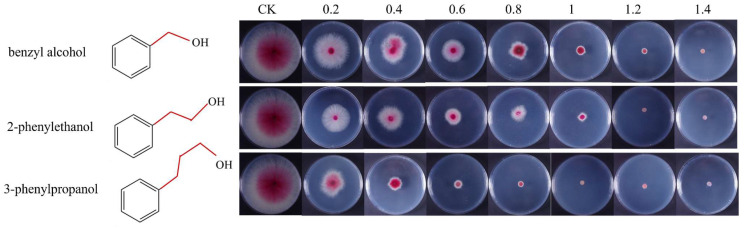
The in vitro antifungal activity of benzyl alcohol, 2-phenylethanol, and 3-phenylpropanol against *F. graminearum*. Mycelia were cultured on PDA medium containing different concentrations of benzyl alcohol, 2-phenylethanol, and 3-phenylpropanol for 3 days.

**Figure 7 microorganisms-11-02954-f007:**
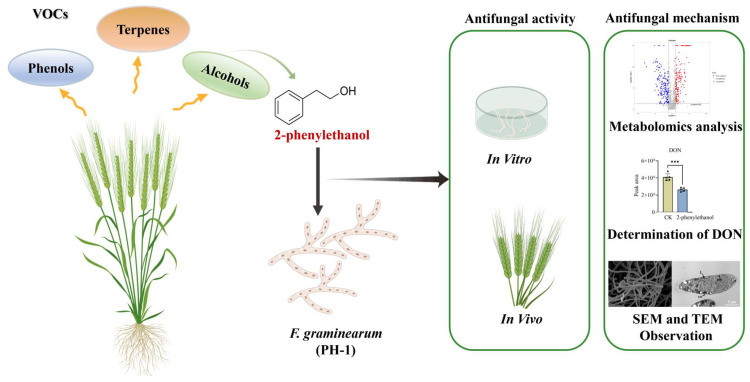
Schematic representation of the potential regulation of 2-phenylethanol in *F. graminearum*.

**Table 1 microorganisms-11-02954-t001:** EC_50_ values of the title compounds against *F. graminearum* (mg/mL).

Compound	Regression Equation	*R* ^2^	EC_50_ (mg/mL)	95% Confidence Interval
benzylalcohol	y = 1.837x − 0.687	0.973	0.374	1.358–1.960
2-phenylethanol	y = 2.20x − 0.071	0.970	0.328	1.898–2.501
3-phenylpropanol	y = 2.341x − 0.514	0.988	0.220	1.809–2.873

## Data Availability

All the data related to this project are presented here.

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
