# Peer review of "Effects of 2-Phenylethanol on Controlling the Development of Fusarium graminearum in Wheat"

_microorganisms, 2023, doi:10.3390/microorganisms11122954_

Round 1
Reviewer 1 Report
Comments and Suggestions for Authors
This is a study on the production of 2-phenylethanol by a wheat cultivar resistant to Fusarium graminearum infection. The topic is interesting and the experimental design is appropriate, however, the presentation and discussion of the data requires significant improvements. My key concerns are listed below, and several comments are made throughout the attached manuscript to assist the authors in the review process:
- I suggest a more general title, as I believe that more studies are needed to prove and better explain damage to the cell membrane and inhibition of mycotoxins production.
- The study used a genetically modified strain of wheat, but the authors do not explain how this modification could interfere with the results obtained.
- Why do some experiments use the EC50 value at 0.328 mg/mL, while others (such as metabolomics and DON accumulation) used 1.2 mg/mL?
- The discussion about the effect of 2-phenylethanol on protein metabolism is very speculative. I believe that the attempt to understand the effects of this component on the inhibition of the fungus must go further. Therefore, I suggest that the authors deepen the discussion based on the available literature and relating it to the data obtained from this study, for example, you could investigate how metabolic pathways involved in the production of mycotoxins and their byproducts are inhibited.
- Lastly, the study of the relationship between antifungal activity and the size of the carbon chain of alcohols was somewhat lost. The topic must be presented in an integrated way with the others, and the data obtained incorporated (as far as possible) into the general understanding of the problem presented in the study.
Methodology:
There are sections of the methodology where the writing resembles a protocol, such as lines 123-125; 139-145; 158. A fluid and consistent text would be much better.
Results:
item 3.1
Figure 1 - The 2-phenylethanol GC-MS chromatogram (1E) displays relative abundance X m/z, while the chromatogram of 2-phenylethanol from experiments are signal strength X time. Is the comparison of these chromatograms correct? It is necessary to better explain this correlation in the text to ensure the credibility of the data and prove the presence of 2-phenylethanol in the experimental condition of the Fig. 1B;
item 3.3
Figure 2 - The description of the results for both in vitro and in vivo assays are insufficient. It is necessary to detail it in the text and point out the images for better understanding. Especially about the in vivo assays, I can't see clear differences in the images, please indicate it. Furthermore, the figure subtitle needs to be revised.
item 3.6
The image analyzes (Figure 5) were a great choice, however, to get even more benefit, the resolution of images D, E and F must be improved. Furthermore, the order of magnitude of images B and E are different, 5 and 2 μm, respectively. The resolution of Figure 3 also needs to be improved;
Comments on the Quality of English LanguageModerate editing are required
Author Response
Dear Editor,
On behalf of all the contributing authors, I would like to express our sincere appreciation of your letter and reviewers’ constructive comments concerning our article entitled “Effects of 2-phenylethanol on Controlling the Development of Fusarium graminearum in Wheat” (ID: microorganisms-2719730). These comments are all valuable and helpful for improving our article. According to the associate Editor and reviewers’ comments, we have extensively modified our manuscript. In this revised version, changes to our manuscript were all shown within the document using red-colored text. Point-by-point responses to the nice Associate Editor and nice reviewers are listed below this letter.
Reviewer #1:
- I suggest a more general title, as I believe that more studies are needed to prove and better explain damage to the cell membrane and inhibition of mycotoxins production.
Response: Thanks for your suggestion. We have revised the title and highlighted it in our manuscript: “Effects of 2-phenylethanol on Controlling the Development of Fusarium graminearum in Wheat”.
- The study used a genetically modified strain of wheat, but the authors do not explain how this modification could interfere with the results obtained.
Response: Thank you for your comments. Line 32-34, We have added to the Introduction. Fhb1, a major gene for resistance to Fusarium head blight (FHB), has been widely used in wheat breeding worldwide for improving FHB resistance [8, 9].
This study used near-isogenic lines of wheat as experimental materials to minimize the variation in volatile compound composition caused by genetic background, enabling precise screening within a specific range.
- Why do some experiments use the EC50 value at 0.328 mg/mL, while others (such as metabolomics and DON accumulation) used 1.2 mg/mL?
Response: Thank you for your question. We have made modifications to section 3.3 in the manuscript. The main reason for the difference in experimental concentrations of fungicides between in vitro and vivo studies is the significant variations in physiological environments, drug metabolism, and absorption mechanisms between the two. In vitro, fungicides are typically applied directly onto sterile culture media, where they come into direct contact with Fusarium graminearum. In contrast, in vivo conditions involve not only overcoming the barrier of biological membranes and penetrating into tissues but also undergoing a series of metabolism and absorption processes.
- The discussion about the effect of 2-phenylethanol on protein metabolism is very speculative. I believe that the attempt to understand the effects of this component on the inhibition of the fungus must go further. Therefore, I suggest that the authors deepen the discussion based on the available literature and relating it to the data obtained from this study, for example, you could investigate how metabolic pathways involved in the production of mycotoxins and their byproducts are inhibited.
Response: We think this is an excellent suggestion. Based on your comments, we further discussed the antifungal mechanism of 2-phenylethanol in the discussion section. The results suggested that 2-phenylethanol interfered with the protein synthe-sis/degradation of F. graminearum and affected its normal growth. Additional-ly,Moreover, amino acids play a crucial role in the synthesis of DON toxins in F. graminearum. L-isoleucine, a branched-chain amino acid, plays three crucial roles in the production of DON toxins in F. graminearum: firstly, it serves as a precursor for the synthesis of DON toxins; secondly, as a co-factor for enzymes, it enhances the activity of enzymes involved in DON synthesis; and finally, it regulates metabolic pathways, influencing the production and utilization of other metabolites [49-51]. Consistent with these results, both non-targeted and targeted metabolomic analyses indicated a sig-nificant inhibition of DON toxin synthesis and a reduction in the accumulation of DON toxins (Figure 3, 4). The inhibition of DON toxin synthesis in F. graminearum by 2-phenylethanol is of significant importance in ensuring human health and enhancing wheat yield and quality.
- Lastly, the study of the relationship between antifungal activity and the size of the carbon chain of alcohols was somewhat lost. The topic must be presented in an integrated way with the others, and the data obtained incorporated (as far as possible) into the general understanding of the problem presented in the study.
Response: Thanks for your suggestion. We have made some changes in the Results 3.7 and disscusion. Line 304-311. To better excavate the potential alcoholic fungicides, we compared the relation-ship between carbon chain length and antifungal activity. Figure 6 displays preliminary results of three alcoholic compounds against the growth of F. graminearum. The concentrations of benzyl alcohol, 2-phenylethanol, and 3-phenylpropanol that completely inhibited the growth of F. graminearum were 1.4, 1.2, and 1.0 mg/mL, respectively. The EC50 values of three alcoholic compounds are different: benzyl alcohol > 2-phenylethanol > 3-phenylpropanol (Table 1). The three compounds have only dif-ferent carbon chain lengths in their structures, with longer ones resulting in higher antifungal activity.
Line 370-376: After confirming the antifungal activity of 2-phenylethanol, we further explored the relationship between the structure of alcohols and their antifungal activity. The antifungal activity of alcohols (benzylalcohol, 2-phenylethanol, and 3-phenylpropanol) is positively correlated with the length of carbon chain. By comparing the antifungal activity of benzylalcohol, 2-phenylethanol, and 3-phenylpropanol, as well as the production cost, we finally determined the most suitable fungicide for application in field production is 2-phenylethanol.
Methodology:
- There are sections of the methodology where the writing resembles a protocol, such as lines 123-125; 139-145; 158. A fluid and consistent text would be much better.
Response: This is an excellent suggestion. We have revised it in the manuscript.
Line 123-125: 2.6. Metabolomics Analysis
2.6.1. Metabolites Extraction
Metabolomics analysis was performed as described in Zhao et al. [30], with minor modifications. A 0.1 g mycelial sample was weighed and mixed with 1.5 mL extraction solution (methanol: water: formic acid, 2:1:1, v/v/v). The mixture was subjected to ultrasound treatment at room temperature for 20 minutes and then rotated at 4°C for 12 hours. Subsequently, it was centrifuged at 13,000 rpm for 5 minutes, and 2 mL of the supernatant was transferred to a new centrifuge tube. The filtrate was concentrated using a rotary evaporator, re-dissolved in 100 μL of 50% methanol, and filtered through a syringe filter (0.22 μm) for analysis preparation. Each treatment consisted of three biological replicates.
Line 139-145: The solvent gradient was set as follows: A, 0.1% of acetic acid in distilled water; B, 0.1% of acetic acid in acetonitrile. The solvent gradient was set as follows: 0-2.5 min; 98–95% A, 2.5-7.5 min; 95-70% A, 7.5- 10.5 min, 70-2% A, 10.5-13 min, 2% A; 2–98% A, 13.1-11.5 min; 98% A, injection volume: 3 µL.
Line158: In negative polarity mode, spray voltage, 2.9 kv; sheath gas, 40; pilot gas, 0; capillary temperature, 350°C. resolution, 17500; microsweep, 1; AGC target 2e5; normalized collision energy, 50.
Results:
item 3.1
- Figure 1-The 2-phenylethanol GC-MS chromatogram (1E) displays relative abundance X m/z, while the chromatogram of 2-phenylethanol from experiments are signal strength X time. Is the comparison of these chromatograms correct? It is necessary to better explain this correlation in the text to ensure the credibility of the data and prove the presence of 2-phenylethanol in the experimental condition of the Fig. 1B;
Response: Thanks for your suggestion. We have made some changes in the Results 3.1. Secondary mass spectrometry is commonly used for qualitative analysis of substances. m/z represents mass-to-charge ratio. Mass-to-charge ratio, also known as m/z, refers to the ratio of the mass and charge of an ion or peak. The m/z can be considered as the relative mass of an ion. Relative abundance refers to the strength of the ion signal corresponding to each m/z ratio.
During the process of mass spectrometry analysis, the substance being detected undergoes ionization within the ion source and then fragments into ions of various mass-to-charge ratios (m/z). The parent ion is formed when a molecule loses an electron upon being bombarded by an electron beam, and its mass should be close to that of the molecule. The daughter ions are the ions generated in the mass spectrometer after the production and fragmentation of the parent ion. These product ions can provide information about the structure of the compound, thus playing a crucial role in determining the molecular formula and structure of the compound. In this study, the molecular weight of 2-phenylethanol is 122.17. In the secondary mass spectrum (Figure 1E), the parent ion was 122, and the daughter ions were 91 and 65. These results match the information of 2-phenylethanol. Figures 1A-D were chromatograms, primarily used to determine the consistency with standard substances in the database based on retention time. Peak area is mainly used for compound quantification. The absence of peaks in Figures 1A, C, and D indicated that the compound was not detected in the corresponding samples.
item 3.3
- Figure 2 - The description of the results for both in vitro and in vivo assays are insufficient. It is necessary to detail it in the text and point out the images for better understanding. Especially about the in vivo assays, I can't see clear differences in the images, please indicate it. Furthermore, the figure subtitle needs to be revised.
Response: Thank you very much for your careful reading of our paper. Our method of representation has caused confusion for you. We have corrected the images in Figure 2. We confirm that all figures are correct now. Later, we provided a detailed supplement to the results of this section.
item 3.6
- The image analyzes (Figure 5) were a great choice, however, to get even more benefit, the resolution of images D, E and F must be improved. Furthermore, the order of magnitude of images B and E are different, 5 and 2 μm, respectively. The resolution of Figure 3 also needs to be improved;
Response: We apologize for the inconvenience. We have re-uploaded the higher resolution pictures of Figure 3. We have rechecked the images of Figure 5 and they were all obtained at the same resolution. In Figure 5, when the magnification of image B was the same as that of images C, E, and F, the mycelium was not fully captured.
Reviewer 2 Report
Comments and Suggestions for Authors
line 186: plase complete the correct name of the substnce "3-phenylpropanol"
line 205: the description of figure 1 is not clear to understand. What exactly is shown? Please check and adjust by the authors.
line 217: chapter 3.3.2:
On what basis was the concentration of 2-phenylethanol determined for the in vitro test series ? Please show and explain by the authors with results.
line 229: the description of graph 2, (B-F) is not clear to understand. What exactly is shown? Please check and adjust by the authors.
line 248, figure 3: The quality of the individual graphs is poor, the labelling is illegible and therefore unacceptable. Please check and adapt by the authors.
line 259: What is "Fielder", Please adaptiert the correct wording cv. Fielder in the whole text.
Figure 4: Please correct the axis labelling of the graph on the right "3-phenylpropanol" or was the same substance tested twice. Please have the authors check and eliminate these repetition errors.
line 293: Please use the correct spelling for the name of 3-phenylpropanol. Please check and adapt by the authors.
line 312: Please delete the duplication of the words "we found...". Please check and adapt by the authors.
in general:
1. Please explain clearly by the authors why hardly any results on 3-phenylpropanol were presented.
2. The discussion does not address the implementation of the results for the practical control of Fusarium graminearum as pathogen of Fusarium head blight and the option for mycotoxin reduction. Please check with the authors and explain this aspect in the discussion.
Comments on the Quality of English Languageno comments
Author Response
Dear Editor,
On behalf of all the contributing authors, I would like to express our sincere appreciation of your letter and reviewers’ constructive comments concerning our article entitled “Effects of 2-phenylethanol on Controlling the Development of Fusarium graminearum in Wheat” (ID: microorganisms-2719730). These comments are all valuable and helpful for improving our article. According to the associate Editor and reviewers’ comments, we have extensively modified our manuscript. In this revised version, changes to our manuscript were all shown within the document using red-colored text. Point-by-point responses to the nice Associate Editor and nice reviewers are listed below this letter.
Reviewer #2:
- line 186: please complete the correct name of the substance "3-phenylpropanol"
Response: Sorry for this mistake. We have corrected it in the manuscript and highlighted it.
- line 205: the description of figure 1 is not clear to understand. What exactly is shown? Please check and adjust by the authors.
Response: We are sorry that our presentation caused you problems, and we have revised the results in conjunction with the suggestions of other reviewers and provide the following explanations. Secondary mass spectrometry is commonly used for qualitative analysis of substances. m/z represents mass-to-charge ratio. Mass-to-charge ratio, also known as m/z, refers to the ratio of the mass and charge of an ion or peak. The m/z can be considered as the relative mass of an ion. Relative abundance refers to the strength of the ion signal corresponding to each m/z ratio.
During the process of mass spectrometry analysis, the substance being detected undergoes ionization within the ion source and then fragments into ions of various mass-to-charge ratios (m/z). The parent ion is formed when a molecule loses an electron upon being bombarded by an electron beam, and its mass should be close to that of the molecule. The daughter ions are the ions generated in the mass spectrometer after the production and fragmentation of the parent ion. These product ions can provide information about the structure of the compound, thus playing a crucial role in determining the molecular formula and structure of the compound. In this study, the molecular weight of 2-phenylethanol is 122.17. In the secondary mass spectrum (Figure 1E), the parent ion was 122, and the daughter ions were 91 and 65. These results match the information of 2-phenylethanol. Figures 1A-D were chromatograms, primarily used to determine the consistency with standard substances in the database based on retention time. Peak area is mainly used for compound quantification. The absence of peaks in Figures 1A, C, and D indicated that the compound was not detected in the corresponding samples.
- line 217: chapter 3.3.2: On what basis was the concentration of 2-phenylethanol determined for the in vitro test series ? Please show and explain by the authors with results.
Response: We sincerely appreciate the valuable comments. We have modified the content of 3.3. The main reason for the difference in experimental concentrations of fungicides between in vitro and vivo studies is the significant variations in physiological environments, drug metabolism, and absorption mechanisms between the two. In vitro, fungicides are typically applied directly onto sterile culture media, where they come into direct contact with Fusarium graminearum. In contrast, in vivo conditions involve not only overcoming the barrier of biological membranes and penetrating into tissues but also undergoing a series of metabolism and absorption processes. Therefore, in order to ensure the effectiveness of fungicides in practical production applications, the concentrations used in in vivo experiments are typically set higher. In this study, In this study, the in vivo concentration of 2-phenylethanol was based on the concentrations screened in in vitro experiments. 2-phenylethanol exhibited better antifungal effects at a concentration of 0.6 mg/mL.
- line 229: the description of graph 2, (B-F) is not clear to understand. What exactly is shown? Please check and adjust by the authors.
- line 248, figure 3: The quality of the individual graphs is poor, the labelling is illegible and therefore unaccel-ptable. Please check and adapt by the authors.
Response: We are sorry to have made such a serious and embarrassing mistake. We have revised it in the figure 3.
- line 259: What is "Fielder", Please adaptiert the correct wording cv. Fielder in the whole text.
Response: We apologize for the confusion our statement caused you. We have rephrased the sentence and corrected such errors throughout the text. Line 259: DON, 3ADON, and 15ADON accumulation of wheat spikes after inoculation with F. graminearum for 14 days.
- Figure 4: Please correct the axis labelling of the graph on the right "3-phenylpropanol" or was the same substance tested twice. Please have the authors check and eliminate these repetition errors.
Response: Thank you for your comments. We have made modifications to Figure 4 in the manuscript.
- line 293: Please use the correct spelling for the name of 3-phenylpropanol. Please check and adapt by the authors.
Response: Thanks for your careful checks. We have modified it in Line 293.
- line 312: Please delete the duplication of the words "we found...". Please check and adapt by the authors.
Response: We feel sorry for our carelessness. We have deleted the repeated part.
In general:
- Please explain clearly by the authors why hardly any results on 3-phenylpropanol were presented.
Response: We are sorry to have made such a serious and embarrassing mistake, and we have made the change in Line 307-318: To better excavate the potential alcoholic fungicides, we compared the relation-ship between carbon chain length and antimicrobialfungal activity. Figure 6 displays preliminary results of three alcoholic compounds against the growth of F. graminearum. The concentrations of benzylalcohol, 2-phenylethanol, and 3-phenylpropanol that completely inhibited the growth of F. graminearum were 1.4, 1.2, and 1.0 mg/mL, re-spectively. The EC50 values of three alcoholic compounds are different: benzyl alcohol > 2-phenylethanol > 3-phenylpropanol (Table 1). The three compounds have only dif-ferent carbon chain lengths in their structures, with longer ones resulting in higher antifungal activity.
Figure 6. The in vitro antifungal activity of benzyl alcohol, 2-phenylethanol, and 3-phenylpropanol against F. graminearum. Mycelial were cultured on the PDA medium containing different concen-trations of benzyl alcohol, 2-phenylethanol, and 3-phenylpropanol for 3 days.
Line 370-376: After confirming the antifungal activity of 2-phenylethanol, we further explored the relationship between the structure of alcohols and their antifungal activity. The antifungal activity of alcohols (benzylalcohol, 2-phenylethanol, and 3-phenylpropanol) is positively correlated with the length of carbon chain. By comparing the antifungal activity of benzylalcohol, 2-phenylethanol, and 3-phenylpropanol, as well as the pro-duction cost, we finally determined the most suitable fungicide for application in field production is 2-phenylethanol.
- The discussion does not address the implementation of the results for the practical control of Fusarium graminearum as pathogen of Fusarium head blight and the option for mycotoxin reduction. Please check with the authors and explain this aspect in the discussion.
Response: This is an excellent suggestion. We further discussed the research results of this part in our discussion. The results suggested that 2-phenylethanol interfered with the protein synthesis/degradation of F. graminearum and affected its normal growth. Additional-ly,Moreover, amino acids play a crucial role in the synthesis of DON toxins in F. graminearum. L-isoleucine, a branched-chain amino acid, plays three crucial roles in the production of DON toxins in F. graminearum: firstly, it serves as a precursor for the synthesis of DON toxins; secondly, as a co-factor for enzymes, it enhances the activity of enzymes involved in DON synthesis; and finally, it regulates metabolic pathways, influencing the production and utilization of other metabolites [49-51]. Consistent with these results, both non-targeted and targeted metabolomic analyses indicated a sig-nificant inhibition of DON toxin synthesis and a reduction in the accumulation of DON toxins (Figure 3, 4). The inhibition of DON toxin synthesis in F. graminearum by 2-phenylethanol is of significant importance in ensuring human health and enhancing wheat yield and quality.